# Strong ice-ocean interaction beneath Shirase Glacier Tongue in East Antarctica

Daisuke Hirano [1,2 ✉], Takeshi Tamura[3,4,5], Kazuya Kusahara[5,6], Kay I. Ohshima[1,2], Keith W. Nicholls [7], Shuki Ushio [3,4], Daisuke Simizu [3], Kazuya Ono [1], Masakazu Fujii[3,4], Yoshifumi Nogi [3,4] & Shigeru Aoki[1]

Mass loss from the Antarctic ice sheet, Earth's largest freshwater reservoir, results directly in global sea-level rise and Southern Ocean freshening. Observational and modeling studies have demonstrated that ice shelf basal melting, resulting from the inflow of warm water onto the Antarctic continental shelf, plays a key role in the ice sheet's mass balance. In recent decades, warm ocean-cryosphere interaction in the Amundsen and Bellingshausen seas has received a great deal of attention. However, except for Totten Ice Shelf, East Antarctic ice shelves typically have cold ice cavities with low basal melt rates. Here we present direct observational evidence of high basal melt rates (7–16 m yr$^{-1}$) beneath an East Antarctic ice shelf, Shirase Glacier Tongue, driven by southward-flowing warm water guided by a deep continuous trough extending to the continental slope. The strength of the alongshore wind controls the thickness of the inflowing warm water layer and the rate of basal melting.

[1] Institute of Low Temperature Science, Hokkaido University, Kita-19 Nishi-8, Kita-ku, Sapporo 060-0819, Japan. [2] Arctic Research Center, Hokkaido University, Kita-21 Nishi-11, Kita-ku, Sapporo 001-0021, Japan. [3] National Institute of Polar Research, 10-3, Midori-cho, Tachikawa 190-8518, Japan. [4] The Graduate University for Advanced Studies, 10-3, Midori-cho, Tachikawa 190-8518, Japan. [5] Antarctic Climate & Ecosystems Cooperative Research Centre, University of Tasmania, Private Bag 80, Hobart, TAS 7001, Australia. [6] Japan Agency for Marine-Earth Science and Technology, 3173-25, Showa-machi, Kanazawa-ku, Yokohama 236-0001, Japan. [7] British Antarctic Survey, Natural Environment Research Council, High Cross, Madingley Road, Cambridge CB3 0ET, UK. ✉email: hirano@lowtem.hokudai.ac.jp

Recent satellite observations have demonstrated accelerated ice flow from the Antarctic ice sheet[1] and significant thinning of the Antarctic ice sheets and shelves[2]. Most mass loss from the Antarctic ice sheet results from iceberg calving and basal melting of its marginal ice shelves, with basal melting contributing more than half of the total[3,4]. As ice shelves have a buttressing effect on seaward glacial flow[5], their thinning or retreat due to increased basal melt will cause an acceleration in ice discharge into the ocean, and subsequent sea-level rise.

Ice shelves with high basal melt rates[4] seem to be commonly found in regions where the Antarctic Circumpolar Current (ACC), transporting warm Circumpolar Deep Water (CDW), approaches the continental slope, or CDW-origin water is transported poleward along the eastern limb of cyclonic gyres. In this context, the Amundsen and Bellingshausen Sea (ABS) ice shelves in West Antarctica are those that are most susceptible to ocean heat flux that results from CDW inflow[6–8]. An increase in basal melting and consequent thinning of ABS ice shelves coupled with increased ice sheet mass loss is principally driven by strengthening of CDW inflows[2,9].

In contrast, East Antarctic continental shelves are primarily occupied by "cold waters"[10], with East Antarctic ice shelves typically having cold ice cavities. An exception is Totten Ice Shelf (TIS) whose continental slope is located near the eastern limb of a cyclonic gyre in the Australian-Antarctic Basin[11,12]. Recent hydrographic observations have revealed relatively warm, modified CDW inflows into TIS cavity[13,14], which could explain basal melt rates (>10 m yr$^{-1}$) comparable with those for the ABS ice shelves[3,4].

A hitherto overlooked hot spot of basal melting in East Antarctica is the Shirase Glacier Tongue (SGT) in Lützow-Holm Bay (LHB), where the continental slope is located near the eastern limb of the Weddell Gyre in the Weddell-Enderby Basin[15] (Fig. 1b). Shirase Glacier, with ice velocity >2 km yr$^{-1}$[16,17], is one of the fastest outlet glaciers in Antarctica and flows into the southern LHB to form the SGT. Among Antarctic ice shelves, SGT has a relatively small area (~821 km$^2$) with relatively high satellite-derived basal melt rates (7.0 ± 2.0 m yr$^{-1}$)[4]. Until recently, heavy land-fast sea ice has prevented shipboard observations even in summer, and so we have no direct hydrographic evidence to determine the effect of the ice tongue on the ocean, or the processes by which the ocean heat gains access to the ice base. In austral autumn 2016, however, large areas of fast ice broke up[18], allowing us to access the bay in January and February 2017, to make the first comprehensive shipboard observations in front of SGT (Fig. 1a; Methods). Here, we first illustrate the direct hydrographic evidence of warm water inflow/meltwater outflow into/from the SGT cavity. Then, we show a causal link between the seasonality of SGT basal melt and the prevailing wind, using a combination of results from the first extensive research cruise, past hydrographic data (see Supplementary Note 1), a coupled ocean–sea ice–ice shelf model, and a time series of directly measured basal melt rates from a site on SGT.

## Results
**Observational evidence on SGT basal melting by warm water.** An important topographic feature in LHB is a deep glacial trough in the center of bay, providing a connection from the shelf break to the SGT ocean cavity (Fig. 1a). It is ~15 km wide, more than 600 m deep, and deepens southward to ~1200–1400 m beneath the northern SGT (Fig. 1a, see also Methods for details of bathymetric data in LHB). Cold (~−1.83 °C), fresh (~34.23), and oxygen-rich (~6.8 ml L$^{-1}$) Winter Water (WW: defined as $T < $ −1.5 °C)[19] is found from the sea surface to 350–400 dbar. Warm (~0.69 °C), saline (~34.66), and oxygen-poor (~4.6 ml L$^{-1}$)

modified Circumpolar Deep Water (mCDW) lies beneath the WW, filling the deep trough to the ice front (Fig. 2a–e). The ocean at the mouth of LHB (Sta.G3) shows a simple two-layer structure consisting of WW and mCDW, which is the same as that observed on the northeast slope (Sta.X31, Fig. 3). This direct observational evidence indicates the presence of undiluted mCDW on the continental shelf in LHB. Compared with the mouth of bay, the mCDW at the ice front is cooler (~0.14 °C), fresher (~34.58), and more oxygen-rich (~5.0 ml L$^{-1}$), indicating modification of mCDW during its journey from shelf break to ice front. However, the mCDW's temperature at the ice front still exceeds the in-situ freezing point by more than 2.7 °C, with the potential to cause strong melting at the base of the SGT. The mCDW therefore represents a large source of ocean heat that would cause strong melting were it to come into contact with SGT's base. Although our CTD profiles did not reach the sea floor (Methods), the absence of dense shelf waters in LHB suggests that near-sea floor mCDW temperatures are comparable to or warmer than the deepest observation at each station.

According to the along-trough distribution in water properties (Fig. 2a–c), toward the ice front, the water above 400 dbar progressively becomes warmer and lower in oxygen content, with a complex intrusive structure that is most obvious near ice front stations (Sta.A3-E2). This indicates that cold and oxygen-rich WW to the north is being altered by the presence of warm and oxygen-poor water originating from mCDW in the deep trough. Above the subsurface layer, the temperature and salinity of the water lie along the mixing line that is characteristic of ice melting into mCDW[20] (Fig. 3). Two types of low-salinity layers ($S < 34$) were observed near the surface (Fig. 2b): one is a ~20 m-thick, relatively warm and oxygen-poor layer at ice front stations (Sta.A3-E2), and the other a cold and oxygen-rich ~40 m-thick layer, observed at the bay mouth (Sta.G3).

**Distribution of glacial meltwater.** Here, we identify the presence of glacial meltwater from relationships between $\delta^{18}O$ (the stable oxygen isotope ratio) and salinity (Fig. 4). Overall, $\delta^{18}O$-salinity relationships at ice front stations (Sta.A2-A6 and E2) are along a line connecting endmembers of mCDW and glacial meltwater. The lowest $\delta^{18}O$ value ~−0.8‰, which is comparable to the surface water in front of Dotson Ice Shelf[21], is found near surface (at 20 dbar, Fig. 4) within the surface low-salinity layer with its relatively warm and oxygen-poor properties (Fig. 2a–c). Among the ice front stations, lower $\delta^{18}O$ values are observed within the trough (Sta.A2-A4). Ice front waters with properties lying along the mCDW-glacial melt line are a result of ocean-SGT interaction; that is, the lower $\delta^{18}O$ waters contain larger glacial meltwater fractions. In contrast, the $\delta^{18}O$-salinity relationships deviate from the mCDW-glacial melt line with distance from the ice front, and $\delta^{18}O$ values near the surface layer are much higher at the bay mouth (~−0.5~−0.4‰).

Salinity and $\delta^{18}O$ can be used as conservative parameters to quantify the respective freshwater contribution from glacial and sea-ice meltwaters[22] (see Methods). The largest glacial meltwater fraction ~2%, comparable with those observed near Pine Island Glacier Ice Shelf[23,24], is found near the sea surface at the ice front stations (Fig. 5b), as estimated from the $\delta^{18}O$-salinity relationships (Fig. 4). The glacial meltwater fractions generally decrease with increasing depth and distance from the SGT ice front. Large glacial meltwater fractions are confined within the surface layer above ~100 dbar, as shown in Fig. 5b. The column-integrated volume of glacial meltwater reaches a maximum at ice front stations (Fig. 5a), as a result of relatively high glacial meltwater fractions extending into the subsurface layer (Fig. 5b).

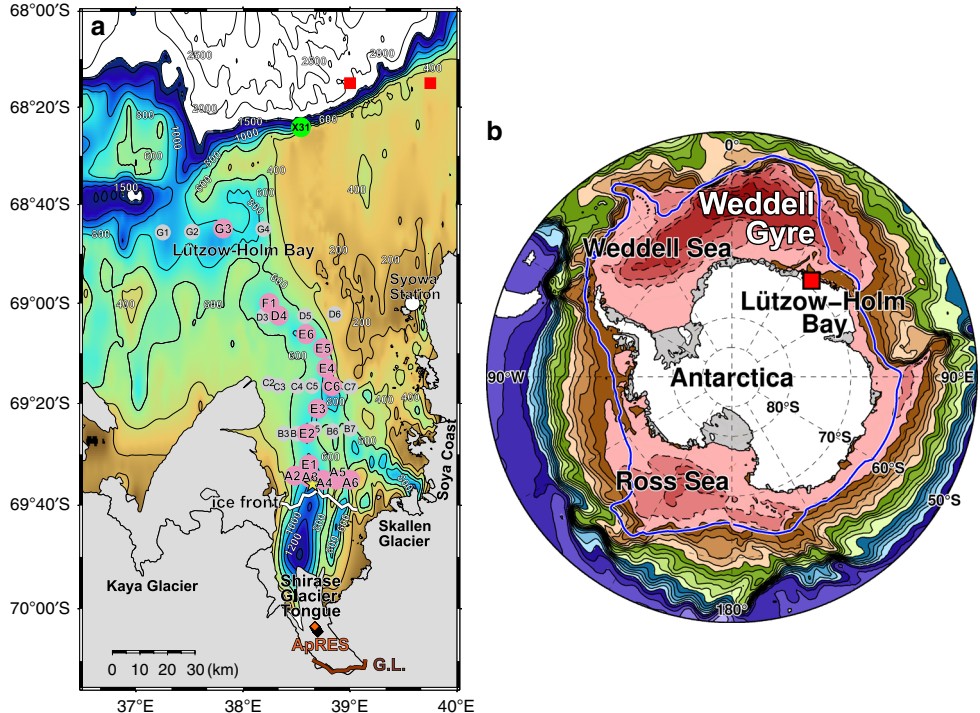

**Fig. 1 Study area of Lützow-Holm Bay, East Antarctica. a** Bathymetry in and around Lützow-Holm Bay, where the domain is indicated by red rectangular in **b**. Pink and gray circles represent positions of CTD stations, and the pink-colored stations are used for vertical sections along (Fig. 2a–c) and across (Fig. 2d, e) the deep trough. Green circle indicates the position of a XCTD cast (Sta.X31) at the northeast continental slope region of the bay. Yellow star near Sta.A3 represents the position of a subsurface mooring with a current profiler, running from late January to August in 1990 (the same position as Sta. P2, see Supplementary Fig. 2). Red squares represent grid locations of wind reanalysis at 10 m height provided by the European Centre for Medium-Range Weather Forecasts (ECMWF) Interim Re-Analysis (ERA-Interim). Orange diamonds indicate positions of ice radar ApRES deployed on the SGT from February 2018 to January 2019. Thick white line shows the SGT ice front derived from MODIS imagery on 22 January 2017, which is different from a model's ice front position (see Supplementary Fig. 6). Thick brown line represents the SGT grounding line[71]. **b** Simulated annual mean vertically integrated transport stream functions[32]. Red colors with dashed contours indicate cyclonic circulation. Contour interval is 10 Sv (1 Sv = 1.0 × 10⁶ m³ s⁻¹). Thick blue line shows the southern boundary of the ACC[6].

The fact that ice-front water properties are relatively warm and oxygen-poor even above the subsurface layer (Fig. 2, Supplementary Fig. 3) is consistent with the high mCDW fractions estimated for these layers from the temperature, salinity, and oxygen profiles (Supplementary Fig. 5b). This suggests a circulation pattern with along-trough mCDW inflow into the SGT cavity, melting and consequent buoyant rise along the SGT base, and then a northward export of this meltwater product (i.e., a glacial meltwater-mCDW mixture) above the subsurface layer (see the next section for further discussion of the meltwater product). Furthermore, the higher glacial meltwater volumes found at the central to western trough stations (Fig. 5c, d) indicate that the glacial meltwater is transported primarily through the western half of the trough after emerging from beneath the SGT. At the bay mouth (Sta.G3), where WW dominates from surface to subsurface layers (Fig. 2a–c, Supplementary Fig. 5c), near-surface sea-ice meltwater fractions (~2.4%, not shown) significantly exceed those of the glacial meltwater (Fig. 5b). We conclude that the low-salinity surface layers are formed from different fresh-water sources: glacial meltwater at the ice front and sea-ice meltwater at the bay mouth.

The temperature profiles obtained from over the deep trough in austral winter (August) and spring (October) in 1990 also show anomalous warm signals within the subsurface layer (200–400 dbar), with the subsurface warm signals gradually diminishing with distance from the ice front (Supplementary Fig. 3). Although mCDW inflow along the deep trough toward the SGT occurs throughout the year, according to the past hydrographic data

from 1990 to 1992 (Fig. 6d, Supplementary Fig. 4), water temperatures above the subsurface layer near the ice front (Sta.P2, Fig. 6d) in summer are markedly higher than in other seasons, with the seasonal temperature variation decreasing with distance from the ice front (Sta.L4 and OW4, Supplementary Fig. 4). In addition, a subsurface mooring near the ice front (300 dbar at Sta. P2) demonstrates that northward flow, up to ~10 cm s⁻¹, mostly dominates throughout the mooring period (late January to August, 1990, Fig. 6b), and subsurface temperatures are higher in summer (~−0.5 °C) and lower in fall (~−1.5 °C) (Fig. 6a, d). It strongly suggests, therefore, that SGT basal melting is year-round, but with a seasonal cycle.

**SGT basal melting with a clear seasonal cycle.** The thickness of the WW layer, or the depth of the thermocline separating WW and mCDW, is closely related to the ocean heat flux into SGT's sub-ice cavity, since the ocean at the mouth of LHB shows a typical two-layer structure, consisting of WW and mCDW (Fig. 2), which is common to the northeast slope region (Fig. 3). The seasonal variation in WW thickness in LHB is caused by seasonality in the prevailing easterly (alongshore) wind over the coastal ocean[19]. The WW layer deepens in fall (typically ~500 m) as a result of enhanced Ekman convergence forced by intensified easterly winds, while the WW layer thickness is decreased (typically 350–400 m) when the easterly wind relaxes in summer. The seasonal variation in prevailing wind is associated with equatorward (in summer) and poleward (in fall) shifts of the atmospheric convergence line around the Antarctic continent[25].

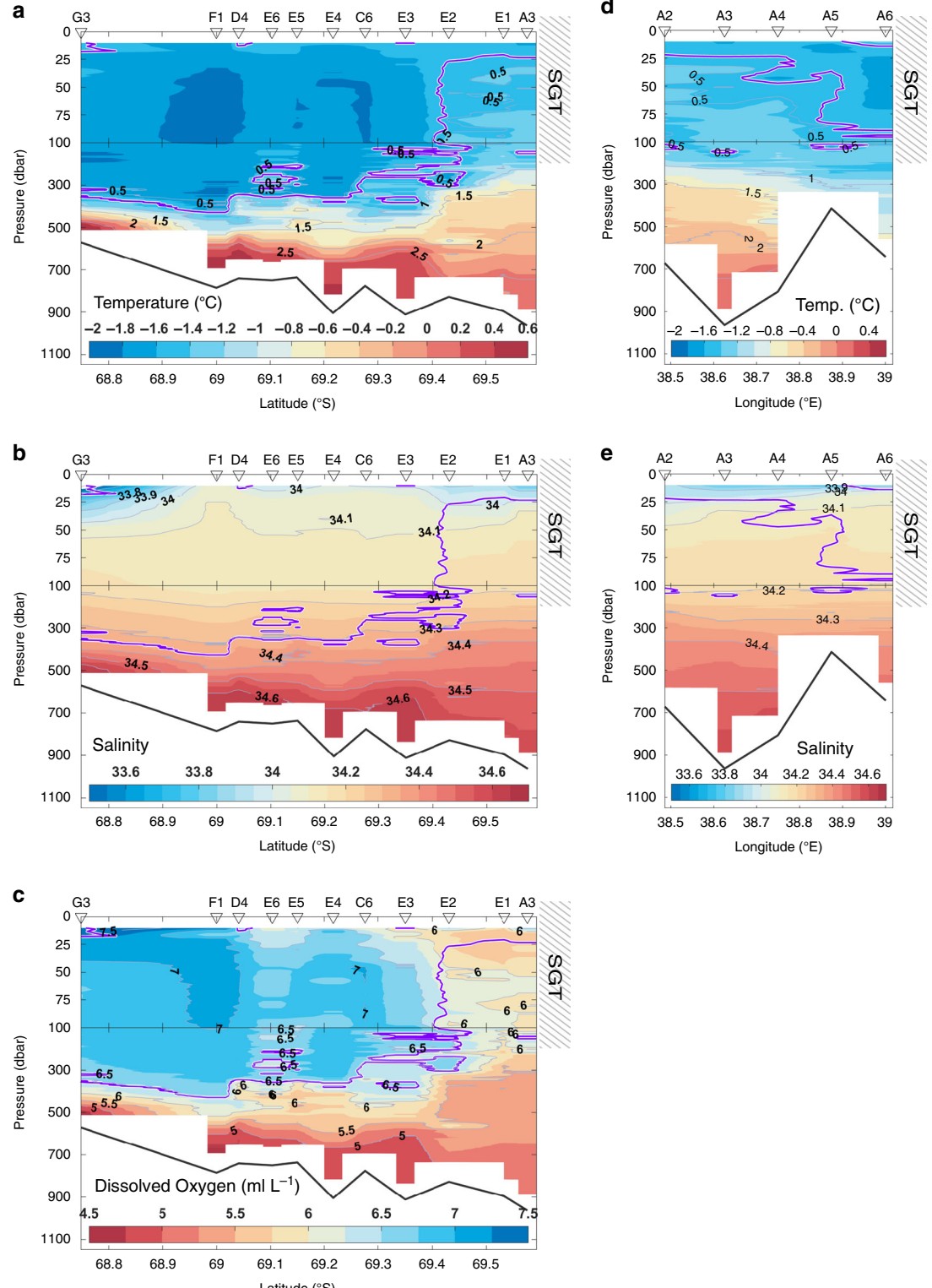

**Fig. 2 Warm water inflow/meltwater outflow into/from SGT cavity.** Along-trough vertical sections of (**a**) temperature (°C), (**b**) salinity, and (**c**) dissolved oxygen (ml L$^{-1}$). Across-trough vertical sections at SGT ice front stations (Sta.A2-A6) of (**d**) temperature (°C) and (**e**) salinity. On panels **a** and **d**, color shading (with color scale) and contours (with contour labels) represent temperature and temperature anomaly from in-situ freezing point, respectively. Contours of $T = -1.5$ °C are shown by thick purple lines (WW layer is defined as $T < -1.5$ °C)[19]. Hatched boxes represent the approximate ice draft ~200 m at the northern part of the SGT. Note that vertical scales above and below 100 dbar are different.

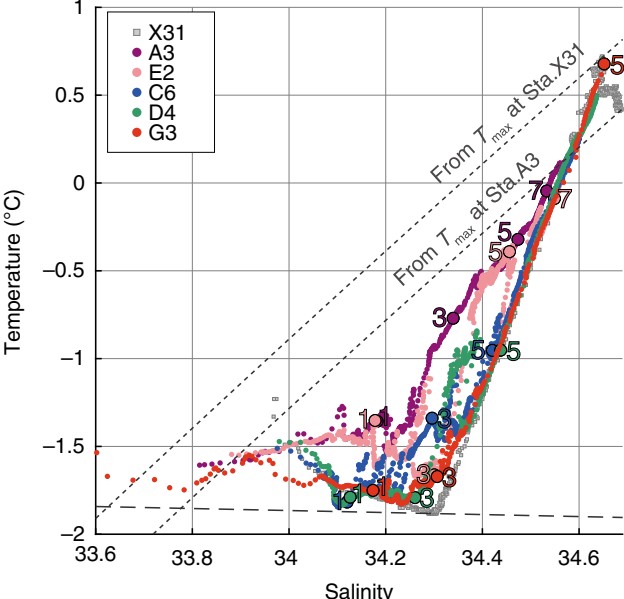

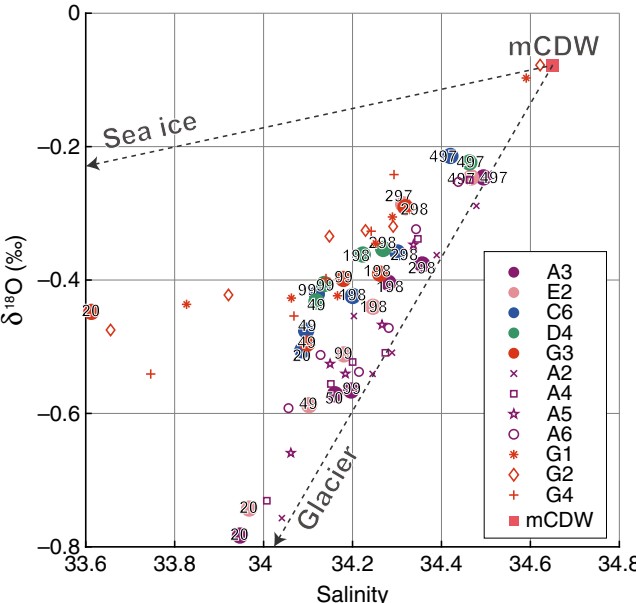

**Fig. 3 Glacial meltwater signals on *T-S* space.** Temperature-salinity relationships at five trough stations (Sta.A3, E2, C6, D4, and G3) with the northeast slope region (Sta.X31, shown by gray symbols). Two dashed diagonals denote theoretical mixing lines ("Gade line")[20] between glacier (ice) and observed mCDW temperature maxima (at Sta.A3 at the SGT ice front and Sta.X31 on the northeast slope region). Horizontal dashed line is the freezing point of seawater at atmospheric pressure. Numbers 1, 3, 5, 7 beside the large symbols indicate pressure levels at 100, 300, 500, and 700 dbar, respectively.

**Fig. 4 Glacial meltwater signals on $\delta^{18}$O-S space.** $\delta^{18}$O (the stable oxygen isotope ratio)-salinity relationships at five trough stations (colored circles with numbers showing the pressure where water samples were obtained), ice front stations (purple symbols), and mouth of LHB (red symbols). Two dashed lines extending from "mCDW" denote lines connecting two endmembers of "mCDW" with "glacier" and "sea ice", respectively. Note that $\delta^{18}$O-salinity relationships for "glacier" and "sea ice" endmembers are beyond the range of plot.

Alongshore wind speed from the ERA-Interim reanalysis on the northeast continental slope (see their grid locations in Fig. 1a) shows a clear seasonal variation, which is intensified in autumn and weakened in summer (Fig. 7a), as previously demonstrated[19]. Such variability in the thermocline depth caused by easterly wind-driven surface Ekman dynamics is common to much of the East Antarctic continental slope, and is one of the essential factors controlling the ocean heat flux beneath the adjacent ice shelves[26–29].

In a fast-flowing outlet glacier such as Shirase Glacier, subglacial meltwater discharged from the glacier bed across the grounding line, along with meltwater produced at the ice-ocean interface, provides a source of buoyancy that can force an overturning circulation along the ice shelf base[30]. This buoyant plume increases its volume flux through entrainment of the surrounding seawater, and the oceanic heat entrained into the plume promotes the melting at the ice-ocean interface[30]. As suggested in the previous section, the northward transport of the mixture of glacial meltwater and mCDW that would result from such a plume would explain the anomalous warm signals observed at the subsurface layer of the SGT ice front (Figs. 2a–c, 3, and 6d, Supplementary Fig. 3). Thus, the northward current velocity and water temperature in the anomalous subsurface warm layer reflects the magnitude of the overturning circulation accompanied by mCDW entrainment along the SGT base. Further, given the assumption that a plume is well-mixed at the ice shelf base, the basal melt rate can be assumed to depend on the product of the temperature above freezing point and the water speed. Here, we propose a "basal melt flux index" (Fig. 6c), simply defined as ($T - T_f$) × $v$, using subsurface mooring data at 300 dbar near the ice front (Sta.P2, Fig. 6a, b), where $T$ is the temperature, $T_f$ the in-situ freezing point, and $v$ the northward velocity. The basal melt flux index peaked in summer, fell in fall, and increased again in winter

(Figs. 6c, 7a). This corresponds well to the variability in basal melt rates derived from the simulation and in-situ ice radar measurement discussed below (Fig. 7a, c). From this, we assume that changes in the index reflect those in the basal melt rate across the entire SGT sub-ice cavity. We note that the seasonal cycle of the basal melt flux index is inversely correlated with that of the wind stress on the northeast continental slope (Fig. 7a). This suggests that, as with much of East Antarctica where easterly winds prevail[26–28], seasonal variability in SGT basal melting is mediated by that of WW thickness, which itself results from the seasonal variability in the easterly wind.

We now use a coupled ocean–sea ice–ice shelf model[31,32] (see Methods) to examine the seasonal variability in SGT basal melting and attempt to provide supporting evidence for its link to variability in alongshore wind stress over the continental slope. The coupled model reproduced the major observed features: a circulation pattern comprising mCDW inflow into the SGT cavity, and northward outflow from beneath it, and the water mass structure and properties along and across the deep trough (Fig. 7b, Supplementary Figs. 6 and 7). The model simulation demonstrates year-round SGT basal melting, with a clear seasonal cycle. Following the pattern in the alongshore wind stress (Fig. 7a), the simulated basal melt rate (annual mean ~9 m yr$^{-1}$) is maximum in summer ~14.5 m yr$^{-1}$ and minimum in fall ~6.0 m yr$^{-1}$, corresponding to the seasonal variability of ocean heat flux from mCDW flowing into the SGT cavity (as shown by the volume flux below $\sigma_\theta = 27.5$ kg m$^{-3}$ in Fig. 7c). The correspondence between the seasonal cycle in basal melt rate (Fig. 7c), the observed basal melt flux index (Fig. 7a) and the WW thickness (Figs. 7c, 6d Supplementary Fig. 4) is particularly strong.

Nearly a complete year of ice radar data (ApRES from February 2018 to January 2019, see Methods) from a site on the SGT strongly supports the seasonality in the SGT basal melting

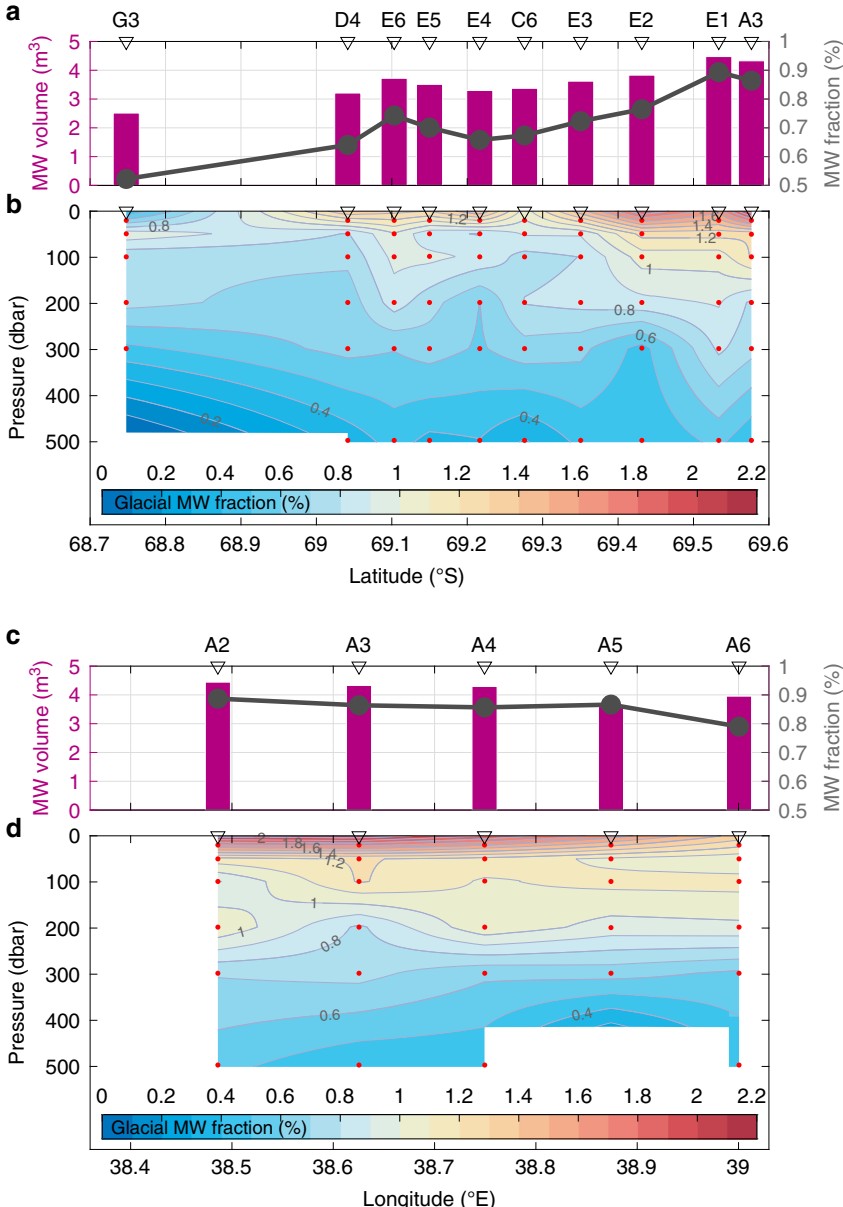

**Fig. 5 Distribution of glacial meltwater.** Column-integrated glacial meltwater volume per unit area (purple bar) and mean glacial meltwater fraction (%, line with dot) above 500 dbar (**a**) along- and (**c**) across-trough stations. Vertical sections of glacial meltwater fraction (%) (**b**) along- and (**d**) across-trough stations. Red dots indicate locations for water samples. The glacial meltwater fractions are linearly interpolated between water sample points and extrapolated from 20 dbar to sea surface. For Sta.G3, the extrapolation is conducted also from the deepest sampling layer to bottom.

deduced from the results of the hydrographic observation and model simulation, although the ice radar measurement was conducted at a single point located ~45 km south of the SGT ice front and ~16 km north of the southernmost SGT grounding line (Fig. 1a). The time series of ApRES-derived basal melt rate shows 15–18 m yr$^{-1}$ in summer and 0-6 m yr$^{-1}$ from autumn to winter (red lines in Fig. 7c), demonstrating a clear seasonal signal in the strength of SGT basal melting. The discontinuity in the ApRES data between January 2019 and February 2018 suggests inter-annual variability in the magnitude of SGT basal melting in summer. In fact, the simulation also shows substantial inter-annual variability in the summer SGT basal melting, and the observed interannual variability by the ApRES is in a range of one standard deviation of the simulated basal melt rate (Fig. 7c). The direct measurement of the basal melt rate helps validate the simulated seasonality of the SGT basal melt and demonstrates a

robust association of seasonal variabilities in SGT basal melt and easterly wind strength on the continental slope.

Here, we estimate SGT's basal melt rate, using across-trough distributions of observed glacial meltwater (Fig. 5c, d) and the climatological simulated January current velocity (Fig. 7b). The western half of the trough (between Sta.A2 and A3) is taken to be the major outflow region for glacial meltwater from beneath the SGT, based on the distributions of glacial meltwater (Fig. 5c, d) and current velocity (Fig. 7b) at the ice front. This estimate of SGT basal melt rate needs to posit that the observed glacial meltwater is concurrently produced only beneath the SGT and is passing beneath the SGT ice front for the first time (see more details in Methods). Although the estimate is sensitive to these assumptions and several estimated parameters (outflow area, velocity, and background level of glacial meltwater), we obtain a basal melt rate of ~25.4 m yr$^{-1}$ in January, which is broadly

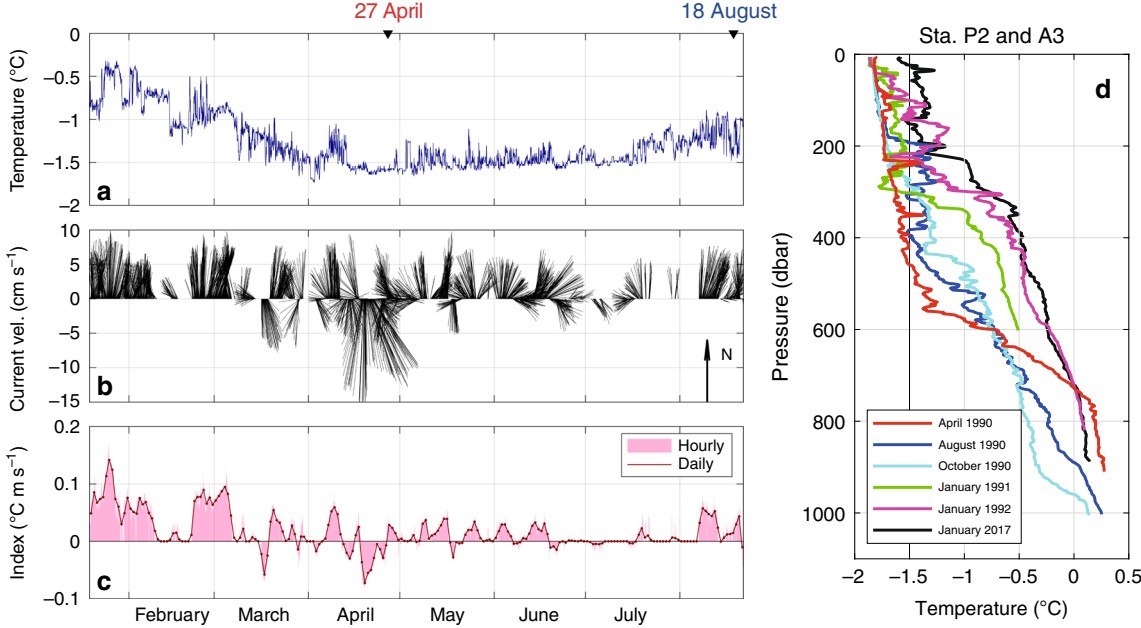

**Fig. 6 Past hydrographic data from 1990 to 1992.** Time series of (**a**) temperature, (**b**) current velocity, and (**c**) basal melt flux index (bar: hourly mean, line with dots: daily mean) at nominal depth of 300 dbar at Sta.P2 (almost the same position as Sta.A3, yellow star in Fig. 1a) from late January to August in 1990. Two inverted triangles on the upper axis of panel **a** denote the date of CTD observations at Sta.P2 (red: April 1990 and blue: August 1990) shown in **d**. **d** Vertical profiles of temperature obtained in April, August, and October 1990, January 1991, January 1992 at Sta.P2 (JARE 31st and JARE 32nd, color), and January 2017 at Sta.A3 (JARE 58th, black). For emphasis, $T = -1.5\,°C$, defined as the lower boundary of Winter Water layer[19], is denoted by vertical thick line.

consistent with simulated (14.5 m yr$^{-1}$) and ApRES-derived (15–18 m yr$^{-1}$) estimates (Fig. 7c). By assuming the simulated seasonal variation in basal melt rates (Fig. 7c), the annual mean basal melt rate is estimated from the January value to be ~16.3 m yr$^{-1}$ (see Methods), which is somewhat larger than but the same order as the value from the simulation (9.3 m yr$^{-1}$, Fig. 7c), ApRES-derived (7.2 m yr$^{-1}$, Fig. 7c), and satellite-derived (7.0 ± 2.0 m yr$^{-1}$)[4] estimates.

The first comprehensive ship-based observations, available past data, wind reanalyses, coupled ocean–sea ice–ice shelf model, and ice radar measurement on the SGT reveal (1) strong basal melting beneath the SGT by the year-round warm water inflows and (2) that the seasonal variation in the prevailing alongshore wind on the continental slope is a key factor controlling the ocean heat flux carried by mCDW into the ice cavity, and, therefore, the magnitude of SGT basal melting.

## Discussion

A striking outcome of this study is the identification of a new hot spot of strong glacial ice-ocean interaction in East Antarctica dominantly driven by warm mCDW inflows (Fig. 8), which is the basal melt process more typical of the ABS ice shelves in West Antarctica. East Antarctic ice shelves, such as Amery and Fimbul Ice Shelves to the east and west of the SGT respectively, typically have cold sub-ice shelf cavities with low basal melt rates (<1 m yr$^{-1}$)[4,27,33], except for Totten Ice Shelf[3,4]. The observed mCDW temperature at the SGT ice front (0.14 °C, Fig. 2a, d) is more than 0.5 °C higher than that at Totten ice front (−0.4 °C)[13]. Hence, the SGT sub-ice shelf cavity is subjected to strong ocean heat forcing, equaling or surpassing TIS, the ice shelf that experiences the highest basal melt rates (10.5 ± 0.5 m yr$^{-1}$)[4] in East Antarctica.

The LHB region (37º–40ºE) is located at the eastern limb of the Weddell Gyre where the southern boundary of the ACC begins to deflect southward (Fig. 1b). We now consider why SGT has a warm sub-ice shelf cavity, atypical of East Antarctica, based on

comparisons with Fimbul Ice Shelf (FIS) in the Eastern Weddell Sea (EWS), located within the same gyre system. A noteworthy similarity, created by the easterly wind-driven Ekman dynamics (poleward transport and consequent convergence), is that the density surfaces of subsurface warm waters (i.e., the thermocline depth) that tilt downward intersect both LHB and EWS continental slopes, preventing cross-slope transport of subsurface warm water onto the continental shelf. This happens at 400–500 m depth at around 40ºE for LHB[19,34], and at 600–700 m depth for EWS[28,35]. Despite this similarity, the inflowing water temperatures are very much lower for the FIS sub-ice shelf cavity (−1.6 °C)[36] than for the SGT cavity (>0 °C, Fig. 2a, d).

The cross-slope transport of subsurface warm water toward the FIS is determined by an eddy-driven overturning and a thermocline depth, that are both modulated by the easterly wind strength[26–28]. While, at LHB the deep continuous trough extending from the northeast continental slope to the SGT (as shown by the 600 m isobaths, indicated with a thick line in Fig. 1a) couples with the shallower thermocline to make LHB much more exposed to off-shelf conditions. Indeed, warm mCDW (>0 °C) flows along the deep trough toward the SGT throughout the year (Figs. 2, 6d, Supplementary Figs. 3 and 4), while maintaining the same two-layer water mass structure over the northeast slope (at Sta.X31, Fig. 3). The continuous deep trough from the continental slope is therefore the key local topographical feature for guiding offshore warm mCDW toward the SGT ice front (Fig. 8). An additional important local factor is the almost permanent cover of heavy land-fast sea ice, which not only dampens the surface Ekman dynamics but also inhibits the formation of a coastal polynya that would otherwise allow intense sea-ice production[37,38] and convection to the sea floor. Accordingly, the inflowing mCDW remains the densest water mass present on the shelf, and fills the trough all the way to the SGT at the southernmost tip of the bay. The continuous deep trough, the permanent cover of land-fast sea ice, and the absence of active

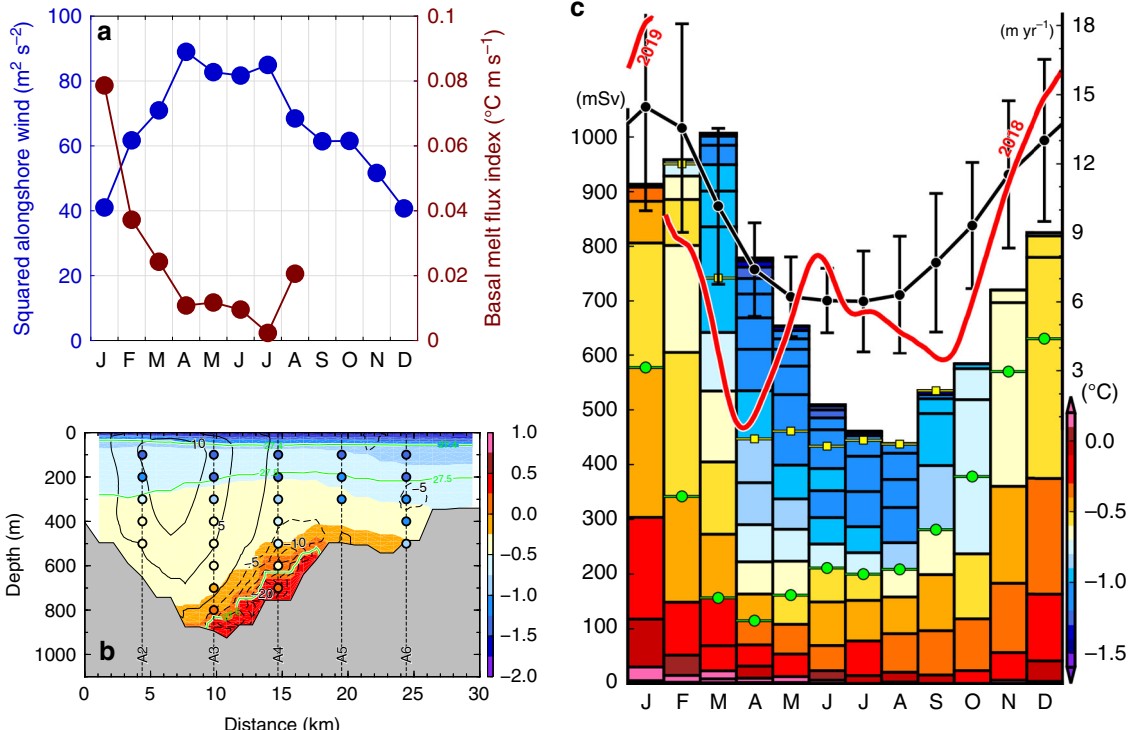

**Fig. 7 Clear seasonal cycle of SGT basal melting. a** Monthly mean basal melt flux index at 300 dbar of Sta.P2 near the ice front (brown, see also Fig. 6c) and monthly mean squared alongshore (southwestward) wind speed averaged for 1979–2017 at the northeast continental slope (blue, equivalent to wind stress). Average of wind speed at two locations (red squares in Fig. 1a) are used here as the representative wind field on the northeast slope region of LHB. **b** Across-trough vertical section of simulated climatology of temperature (color) and northward current velocity (broken contours represent southward current velocity) in January (averaged for 2006–2017, results from the simulation without fast ice: NOFI case, see Methods). Each color of dot denotes observed temperature in January 2017 (see also Fig. 2d). Green contours indicate $\sigma_\theta = 27.4, 27.5$ and $27.6$ kg m$^{-3}$. **c** Simulated seasonal cycle of SGT basal melt rate with a range of one standard deviation (black line, right axis) and water mass flowing into the cavity (boxes, left axis) across the SGT ice front, by the coupled ocean–sea ice–ice shelf model. Color of boxes denote temperature (°C). The volume flux is integrated every 0.02 kg m$^{-3}$, and horizontal yellow and green lines with dot on each month represent $\sigma_\theta = 27.4$ and $27.5$ kg m$^{-3}$, respectively. Time series of ApRES-derived basal melt rate (from February 2018 to January 2019 for 354 days, see Methods) after applying a 36-h low-pass filter is also superimposed (red line, right axis).

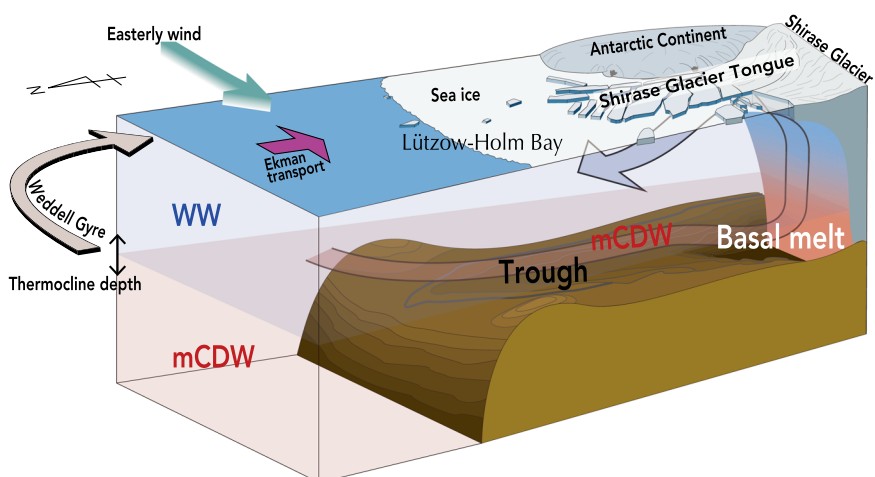

**Fig. 8 Interaction between ocean and SGT, East Antarctica.** The interaction (SGT basal melt process) comprises (1) transport of offshore-origin warm modified Circumpolar Deep Water (mCDW) into Lützow-Holm Bay located in the southeastern part of cyclonic Weddell Gyre, (2) southward inflow of warm mCDW along deep layer of the trough and finally into the region beneath SGT, (3) SGT basal melt by inflowing warm mCDW and buoyant meltwater plume of glacial meltwater together with mCDW, and (4) northwestward outflow of a mixture of glacial meltwater and mCDW above subsurface layer from beneath SGT. Alongshore easterly wind, mediated by surface Ekman dynamics, fluctuates the depth of the thermocline between Winter Water (WW) and mCDW, that is, controls the thickness of warm water inflow into the bay and the magnitude of subsequent basal melting beneath the SGT.

coastal polynyas in LHB provide favorable settings for strong SGT basal melting.

A positive trend in the Southern Annular Mode (SAM) is associated with a strengthening and poleward shift of the Southern Ocean westerly winds[39], which is projected to persist due to continued anthropogenic forcing[40,41]. Under such climatic change, the poleward wind shift decreases the Ekman transport toward the coast and the associated Ekman convergence around the shelf break, increasing ocean heat content at intermediate depths corresponding to the CDW layer[39,42]. Furthermore, positive SAM phases tend to intensify the Weddell Gyre circulation[43], presumably promoting poleward ocean heat transport in the LHB region, and resulting in an intensification of SGT basal melting. LHB is therefore a location that is well suited for monitoring the linkage between the regional winds, the Southern Ocean, and the Antarctic Ice Sheet in a changing climate.

## Methods

**Bathymetric data in Lützow-Holm Bay**. The bathymetry data from ship-based multibeam echo sounders, point echo sounding via sea-ice drill holes, hydrographic survey-based nautical charts made by the Japan Coast Guard (JCG), and the global relief model ETOPO1 were used when making the bathymetric grid in this study.

Multibeam bathymetric data were acquired by a 20 kHz SeaBeam3020 system (L3 Communications ELAC Nautik) installed on the Japanese icebreaker *Shirase*. The sounder has 205 beams with a transducer of 2 (transmission) by 2 (receiving) degrees. The data were acquired during following five expeditions: the Japanese Antarctic Research Expeditions (JARE) 51st during December 2009 to February 2010, JARE 52nd during December 2010 to February 2011, JARE 53rd during December 2011 to March 2012, JARE 54th during December 2012 to February 2013, and JARE 55th during December 2013 to February 2014 (Supplementary Fig. 1a). The data was provided by JCG for the utilization of scientific purpose. The sound velocity correction was conducted by using real-time data from the surface water velocity meter and the sound velocity profiles collected by conductivity, temperature, and depth (CTD), and expendable CTD (XCTD) observations. The HIPS and SIPS software (CARIS, Ltd., Fredericton, Canada) was used for data processing to remove extreme depth variations mainly from the outer edges of the multibeam swaths.

The point echo sounding data via sea-ice drill holes were acquired in winter seasons during the JARE 9th in 1968 to JARE 22nd in 1981 (Supplementary Fig. 1b; e.g., ref. [44]). The measurements were carried out using echo sounders of the GS-3 (28.5 kHz)[45] and NSL-1300 (20 kHz)[46]. Based on results of repeated measurements at the same position, difference of measured depth values from each instrument was less than less 15 meters for ~600-meters-deep seafloor[44]. The survey lines were basically set from east toward west with 1-km intervals. This operation was conducted by using two or more flags and interval distance was measured by the distancemeter of the snow vehicle. Each position was checked up by surveying technique such as triangulation, and resulted error in location of sounding stations was less than 1 km in the greater part of the survey area[44]. These data were compiled into the submarine topographic map of Lützow-Holm Bay and published[47].

The data from the nautical chart created by JCG were mainly used for the offshore region in this study (Supplementary Fig. 1b). They consist of the above-mentioned point echo sounding data as well as ship-based single-beam echo sounding data, which were obtained in Lützow-Holm Bay during the JARE 29th (1988), 33rd (1992), 34th (1993), 37th (1996), and 39th (1998).

Data gaps were filled using the global relief model ETOPO1, in which estimated seafloor topography derived from satellite altimetry mainly covers the area of Lützow-Holm Bay (Supplementary Fig. 1c)[48].

The 1-km resolution bathymetric map was generated using all of the data mentioned above to obtain the final digital terrain model for physical oceanographic modeling in this study (Supplementary Fig. 1d). Data were processed using the GMT system[49], and a bathymetric data was gridded using a weighted nearest-neighbor algorithm and surface algorithm in which adjustable tension continuous curvature splines were used.

**Comprehensive shipboard observations in Lützow-Holm Bay**. From mid-January to early-February (austral summer) in 2017, we carried out comprehensive ship-based hydrographic observations at 31 stations in Lützow-Holm Bay (LHB, Fig. 1a) from the Japanese icebreaker "Shirase" (AGB-5003). This observational campaign was conducted during the 58th Japanese Antarctic Research Expedition (JARE 58th), under the ROBOTICA (Research of Ocean-ice BOundary InTeraction and Change around Antarctica) project of National Institute of Polar Research. The distribution of observation stations within LHB (Fig. 1a) was designed to capture the essential features of off-shelf-originated warm water inflows into the sub-ice cavity of Shirase Glacier Tongue, and meltwater-rich outflows from the cavity. At each station, we measured temperature, conductivity, pressure, and dissolved

oxygen (DO) from the sea surface to 53–98 m above the sea floor, using a conductivity-temperature-depth profiler (CTD; Sea-Bird Electronics SBE19) with a DO sensor (SBE43). Water samples were taken for $\delta^{18}O$ (the stable oxygen isotope ratio) analysis, which were processed at the laboratory of Institute of Low Temperature Science, Hokkaido University in Japan. Water samples were nominally taken at 20, 50, 100, 200, 300, and 500 dbar. In addition, an expendable CTD probe (XCTD; The Tsurumi-Seiki Co., LTD) was deployed on the northeast slope region of LHB (Sta.X31, Fig. 1a).

**Quantification of freshwater sources[22]**. Salinity and oxygen isotope ratio $\delta^{18}O$ can be used as conservative parameters to quantify the respective freshwater contribution from glacial and sea-ice meltwaters. Assuming that observed salinity and $\delta^{18}O$ ($S_{obs}$ and $\delta^{18}O_{obs}$) can be explained as a mixture of three endmembers of modified Circumpolar Deep Water (mCDW), meteoric water (i.e., glacial meltwater), and sea-ice meltwater, the three-component mass balance is described by:

$$f_{sim} + f_{met} + f_{mCDW} = 1, \quad (1)$$

$$f_{sim}S_{sim} + f_{met}S_{met} + f_{mCDW}S_{mCDW} = S_{obs}, \quad (2)$$

$$f_{sim}\delta^{18}O_{sim} + f_{met}\delta^{18}O_{met} + f_{mCDW}\delta^{18}O_{mCDW} = \delta^{18}O_{obs}, \quad (3)$$

where $f_{sim}$, $f_{met}$, and $f_{mCDW}$ are the fractions of endmembers of sea-ice meltwater, meteoric water, and mCDW, respectively. In this study, we used the following endmember values: sea-ice meltwater ($S_{sim} = 0.50$, $\delta^{18}O_{sim} = -5.0‰$, e.g., ref. [50]), meteoric water ($S_{met} = 0.0$, $\delta^{18}O_{met} = -40‰$, e.g., ref. [35]), and mCDW ($S_{mCDW} = 34.65$, $\delta^{18}O_{mCDW} = -0.078‰$) observed at the mouth of LHB (Sta.G3).

**Estimation of glacial meltwater fraction from CTD profiles[51,52]**. Assuming that the observed water mass is explained as a mixture of three endmembers of modified Circumpolar Deep Water (mCDW), Winter Water (WW), and glacial meltwater, the three-component mass balance is described as follows:

$$\chi_{obs}^1 = (1 - \varphi_{MW} - \varphi_{WW})\chi_{mCDW}^1 + \varphi_{MW}\chi_{MW}^1 + \varphi_{WW}\chi_{WW}^1, \quad (4)$$

$$\chi_{obs}^2 = (1 - \varphi_{MW} - \varphi_{WW})\chi_{mCDW}^2 + \varphi_{MW}\chi_{MW}^2 + \varphi_{WW}\chi_{WW}^2, \quad (5)$$

where $\varphi_{MW}$, $\varphi_{WW}$, and $\varphi_{mCDW}(= 1 - \varphi_{MW} - \varphi_{WW})$ are fractions of glacial meltwater, WW, and mCDW, respectively and $\chi$ is a conservative parameter. By solving for $\varphi_{MW}$, we obtain glacial meltwater fraction as

$$\varphi_{MW} = \frac{(\chi_{obs}^2 - \chi_{mCDW}^2) - \alpha(\chi_{obs}^1 - \chi_{mCDW}^1)}{(\chi_{MW}^2 - \chi_{mCDW}^2) - \alpha(\chi_{MW}^1 - \chi_{mCDW}^1)}, \quad (6)$$

where $\alpha = \frac{\chi_{WW}^2 - \chi_{mCDW}^2}{\chi_{WW}^1 - \chi_{mCDW}^1}$. Profile of $\varphi_{MW}$ can be obtained from three pairs of $\chi^1$ and $\chi^2$ (i.e., T and S, DO and T, and DO and S) using CTD-measured temperature, salinity, and DO (dissolved oxygen) as conservative parameters (Supplementary Fig. 5a). Likewise, profiles of $\varphi_{mCDW}$ and $\varphi_{WW}$ are also obtained (Supplementary Figs. 5b and c). In this paper, we set values of three endmembers as follows; mCDW ($T = 0.72\,°C$, $S = 34.65$, $DO = 4.5\,ml\,L^{-1}$), WW ($T = -1.87\,°C$, $S = 34.30$, $DO = 7.0\,ml\,L^{-1}$), and glacial meltwater ($T = -90.0\,°C$, $S = 0$, $DO = 23.4\,ml\,L^{-1}$). Representative values observed at the regions of northeast continental slope and bay mouth are adopted as values for endmembers of mCDW and WW. Temperature of pure glacial meltwater is the intercept of the extrapolated Gade Line with $S = 0$ (Fig. 3). In addition, dissolved oxygen of glacial meltwater is estimated using the empirical relationship between an air content of ice and an elevation at which the ice is formed[53]. Dissolved oxygen of pure glacial meltwater is estimated to be ~23.4 ml L[−1] using the elevation of Shirase Glacier's catchment basin (Mizuho Plateau, ~1500–2000 m)[54].

**Estimation of SGT basal melt rate based on hydrographic data**. With the results from in-situ hydrographic observations and a coupled ocean–sea ice–ice shelf model, we estimate a basal melt rate of SGT. This estimate needs to posit the premise that all the observed glacial meltwaters at the ice front (Fig. 5c, d) are (1) concurrently produced beneath the SGT and (2) passing through the SGT ice front without any recirculation. In fact, however, the observed glacial meltwaters might include some that has recirculated or has been supplied from neighboring glaciers such as Kaya and Skallen Glaciers for example (see their location in Fig. 1a). Since relatively large meltwater fractions ~0.4–0.6% are identified even below the subsurface layer at the trough stations (Fig. 5b, d), where mCDW dominates (Fig. 2, Supplementary Fig. 5b), we assume that ~0.5% (mean meltwater fraction from 400 to 500 dbar at Sta.A2-A4) of the observed meltwater at the ice front comes from recirculation or from other glaciers. Thus, for the estimate of SGT basal melt rates, we treat 0.5% of the observed meltwater fraction as a background level.

Given that the SGT meltwaters are mostly passing through the water column shallower than 300 m in the western half of the trough (Sta.A2-A3, i.e., the area of outflow region is $300 \times 5000\,m^2$), we obtain a glacial meltwater transport of ~20.9 Gt yr[−1] when we adopt mean January values of the observed glacial meltwater fraction of 1.06% (Fig. 5d), and the simulated northward velocity is 0.08 m s[−1] (Fig. 7b). Dividing the meltwater transport by SGT area (~821 km²)[4] yields a basal melt rate of ~25.4 m yr[−1] for January. By assuming the simulated

seasonal variation in basal melt rate (maximum in January ~14.5 m yr$^{-1}$ and annual mean ~9.3 m yr$^{-1}$, Fig. 7c), we calculate an annual mean value of $25.4 \times (9.3/14.5) = 16.3$ m yr$^{-1}$. The estimate is sensitive to the above assumptions about the outflow area, velocity, and the background level of glacial meltwater at the ice front and should therefore be treated with caution.

**A coupled ocean–sea ice–ice shelf model**. This study used a coupled ocean–sea ice–ice shelf model[31,32]. The model used an orthogonal, curvilinear, horizontal coordinate system. Two singular points of the horizontal curvilinear coordinate were placed on the East Antarctic Ice Sheet (72°S, 30°E and 69°S, 50°E) to regionally enhance the horizontal resolution around the Lützow-Holm Bay (LHB) region, while keeping the model domain circumpolar Southern Ocean with the artificial northern boundary at around 30°S.

The vertical coordinate system of the ocean model was z coordinate. The vertical grid spacing was 5 m (4 grid levels) and 20 m (49 grid levels) for 0–20 m and 20–1000 m depth range, respectively. In 1000–2000 m/2000–3000 m/ 3000–5000 m range, we used 20/10/10 grid levels with a spacing of 50/100/200 m. The maximum ocean depth in the model was set to 5000 m to save on computational resources. A partial step representation was adopted for both the bottom topography and ice shelf draft to represent them optimally in the z-coordinate ocean model[55]. The sea ice component used one-layer thermodynamics[56] and a two-category ice thickness representation[57]. Prognostic equations for momentum, mass, and concentration were taken from Mellor and Kantha[58]. Internal ice stress was formulated by the elastic-viscous-plastic rheology[59] and sea ice salinity was fixed at 5 psu.

In the ice shelf component, we assumed a steady shape in the horizontal and vertical directions. The freshwater flux at the base of ice shelves was calculated with a three equation scheme, based on a pressure-dependent freezing point equation and conservation equations for heat and salinity[60,61]. This model did not include tidal forcing, and thus we used the velocity-independent coefficients for the thermal and salinity exchange velocities (i.e., $\gamma_t = 1.0 \times 10^{-4}$ m s$^{-1}$ and $\gamma_s = 5.05 \times 10^{-7}$ m s$^{-1}$)[60]. The modeled meltwater flux and the associated heat flux were imposed on the ice shelf–ocean interface. Note that the modeled basal melt rate and amount depend on the selected values of the coefficients[61]. Estimates of basal melt rate from the oceanographic and ice radar measurements support that our choice of the parameters is reasonable in this modeling framework.

The horizontal grid spacing over the LHB region was less than 2.5 km. This relatively high horizontal resolution enabled us to produce a realistic coastline and bottom topography. The bathymetry for the Southern Ocean in this model was derived from the ETOPO1[62], while ice shelf draft and bathymetry under the ice shelf were obtained from the 1-min refined topography (RTopo-2) dataset[63]. The bottom topography in the LHB region (35–40°E and 68–70°S) was replaced with a detailed topography that blended multi-beam survey in JARE 51st–55th, depth information at control points from the Japan Coast Guard, and ETOPO1 (see Methods for details of the acquisition and processing of bathymetric data used in this model). Fast ice is sea ice that is fastened to the Antarctic coastline and the edges of ice shelves. Extensive fast ice has been identified along the East Antarctic coast[37,64]. We introduced areas of multiyear fast ice into the model as constant-thickness (5 m) ice shelf grid cells. Although in reality the horizontal distribution and thickness of fast ice vary seasonally and interannually[18,64,65], as a first approximation the spatial distribution of multiyear fast ice in the model was assumed to be constant.

North of 40°S, temperature and salinity were restored to the monthly mean climatology of the World Ocean Database[66] throughout the water column with a damping time scale of 10 days. Outside of the focal region the horizontal resolution becomes coarser than 10 km, and sea surface salinity was restored to the monthly mean climatology to suppress unrealistic deep convection in some regions (e.g., Weddell Sea). Daily surface boundary conditions for the model were surface winds, air temperature, specific humidity, downward shortwave, downward longwave, and freshwater flux. To calculate the wind stress and sensible and latent heat fluxes, we used the bulk formula[67]. When the surface air temperature was below 0 °C, precipitation was treated as snow. Daily reanalysis atmospheric conditions were calculated from the ERA-Interim dataset[68]. The ice-ocean model was first integrated with fast ice for 20 years using 2005 forcing, and then hindcast simulations with and without fast ice (FI and NOFI cases) were carried out for the period of 2006–2017 with interannually varying forcing. In this study we utilized the model results from the NOFI case, as the observations were carried out when the fast ice was absent.

**Basal melt rate time series from ApRES**. The autonomous phase-sensitive radio echo sounder (ApRES) is an active radar that was deployed on Shirase Glacier Tongue. It uses the frequency modulated continuous wave approach, transmitting a tone that scans from 200 to 400 MHz over a period of 1 s, at an output power of 100 mW. The instrument measures the change in distance between the radar antennas and the ice base, assuming an appropriate speed of radio waves in ice, and, after appropriate processing, the data yield a time series of basal melt rates[69].

The ApRES dataset covered a period of 354 days from February 2, 2018. Once every hour the instrument collected a burst of 20 measurements over a period of about 21 s. To process the data, each burst was averaged, and then its Fourier transform calculated using the methodology described by Brennan et al.[70]. The

result is a sequence of radar returns, which retain both the phase and amplitude of the signal.

Each return shows a strong reflecting horizon at a depth of about 500 m, indicating the depth of the ice base below the surface, assuming a dielectric permittivity of 3.18 throughout the ice column. The accurate depth of the ice base is not required here, but the phase sensitivity of the measurement means that the vertical motion of the base with respect to the radar antennas can be monitored with a precision that formally depends on the signal to noise ratio of the signal: a high signal to noise ratio of 60 dB, as in the case of the bed reflection in the present study, yields a range precision of less than $10^{-7}$ m. However, the thickness variation is the result of a combination of several effects: basal melting, strain thinning in the ice column, vertical compaction in the firn, sinking of the antennas in the snow, and the temperature sensitivity of instrument itself. The accuracy of the melt rate estimates is therefore set by the ability to account for these other effects.

A key assumption is that the shape of the ice base does not change with time. This is not the case at the Shirase site, and so some care was needed to reduce the effect of the time-variation of the shape of the basal echo on the extracted ice thickness time series. This was achieved by calculating the phase of the cross correlation between the first ten meters of each basal return and the subsequent one. Although the details of the resulting time series is sensitive to which ten-meter portion of basal echo is used, the broad pattern was robust.

By assuming that internal reflecting horizons from within the ice column are fixed in the ice, we use their changes in range to determine the non-melt induced contributions to the thickness change, including the apparent contribution from the temperature-induced variations in the instrument. In principle, this allows us to find the thickness change contribution due to basal melting. As the strength of internal reflections is relatively low, their phase is known with less precision, and short-term (diurnal and faster), non-melt-induced variations in thickness are difficult to extract. This is important for tidally-induced vertical strain, which is expected to make a strong contribution. For that reason, we filter the signal to remove variability at timescales of 36 h and shorter. The final result is a basal melt rate time series uncontaminated by other factors affecting the ice shelf thickness. As a result of the effects discussed above, the estimated error of the derived melt rate is 0.5 m yr$^{-1}$.

## Data availability
The observational and simulation data/results that support the findings of this study are available from the corresponding author upon reasonable request. The ERA-Interim data were obtained from the ECMWF Research Data Server (http://data.ecmwf.int/data/).

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

## Acknowledgements

We are deeply grateful to the officers, crews and scientists on board icebreaker *Shirase* for their help with field observations. Part of the bathymetric data in Lützow-Holm Bay were provided by the Japan Coast Guard. The model simulation was carried out using the Fujitsu PRIMERGY CX600M1/CX1640M1 (Oakforest-PACS) in the Information Technology Center, The University of Tokyo. This work was supported by Grants-in-Aids for Scientific Research (JP17K12811, JP17H01615, JP25241001, JP17H01157, JP17H06316, JP17H06317, JP17H06322, JP17H06323, JP17H04710, JP26740007, JP19K12301, and JP20K12132) of the Ministry of Education, Culture, Sports, Science and Technology, the Science Program of Japanese Antarctic Research Expedition (JARE) as Prioritized Research Project, National Institute of Polar Research (NIPR) through Project Research KP-303, the Center for the Promotion of Integrated Sciences of SOKENDAI, and the Joint Research Program of the Institute of Low Temperature Science, Hokkaido University.

## Author contributions

D.H. conceived this study. D.H. and T.T. designed the field observations. T.T. led the shipboard observations. K.O. and D.S. conducted the observations. K.I.O. and S.U. conducted the on-ice hydrographic observations. K.K. performed the numerical simulation. K.W.N. processed the ice radar data. M.F. processed the bathymetric data. Y.N. contributed to bathymetry data acquisition. S.A. led the Science Program of Japanese Antarctic Research Expedition (JARE) as Prioritized Research Project (ROBOTICA project) and measured the sampled water. All authors discussed the results and commented on the manuscript.

## Competing interests

The authors declare no competing interests.
