## [Peer Review File · Nature Communications]

Reviewers' comments:

Reviewer #1 (Remarks to the Author):

Antarctica may melt substantially in the future, and contribute largely to future sea level rise. So new melt rates and processes in an area that has not yet been properly observed is indeed interesting for a large audience.

The dynamics that explain the 1 year of seasonal variability is consistent with earlier studies along the larger East Antarctic ice shelf areas. But only comparison is made to the Amery ice shelf to the east. Comparison need also to be done to the Fimbul ice shelf to the west, that is actually more comparable – at least when it comes to the bedrock topography. I also found the “meltwater outflow” index needs both better description, and analysis, before it can be meaningfully applied.

There are some, but not many specific comments below. These go in detail on the larger issues noted above. Otherwise this paper both presents observations, simulations and re-analysis in a clear and well documented story – so I recommend publication after these minor revisions are done.

Specific comments:

Generally nice and clear text: “the” is missing in a few places; Line 85 and 156; “the ice front”, Line 201; “the January value”. Figure 1 caption (Line 475) “the deep through”. Line 540 (Fig. 6 caption) “the thermocline”.

Re-analysed wind are essentially a model result, and you need to make the distinction more clear. Line 152: Add: “Alongshore wind speed from re-analysis on the northeast continental slope ...” It is now written as this is observations. Also change from “wind data” to “wind re-analysis” in Fig. 1 caption (Line 479).

Line 118 – 120 is unclear. I suggest re-writing to: “This suggest a circulation pattern with inflow in the sub-surface layer, melting and related bouyant rise below the SGT, and then a northward export of this melt-water product in the surface layer.”

Line 142-154: In this section only work in this specific part is cited, and tend to be of Japanese authors. The wind driven Ekman transport of the fresher surface layer towards the coast is the same for most of the East Antarctic ice shelves. The paper should at least cite relevant work on the Amery and Fimbul Ice shelf.

Line 161 – “Meltwater Outflow Index”. This is not a very clear index. If there is more melting, given the same amount of mixing and flow, the temperature would go down – not up. So here you are assuming that there is more melt when there is also more entrainment – producing the higher temperature. Your assumptions need to be made clear, and you should test if it would be better with simply using the volume of the melt product only – and leaving the temperature out. What about the thickness of the outflow? This will normally change substantially in an ISW plume (Jenkins 2011). It would also be interesting to learn how much of the variability (Fig. 5 a) is due to speed and temperature.

Line 166-167: Again this is consistent with earlier results along East Antarctica. Other earlier studies MUST be cited here...

Line 198-200: The differences between the calculated (25.4 m/yr) and simulated (14.5 m/yr) is quite large, this is OK – but a reasoning around the difference should be made. What is the least robust part of the assumption? Also the year-to-year variability should be discussed shortly. This is clear from the discontinuity in Gig. 5 c) in the ApRES data.

Line 202: should probably be; "the value of simulated" – not the range of simulated

Line 204; add re-analysis to the data sets used.

Line 214. Here your results should also be compared with the available melt rates from the Fimbul Ice shelf (Hattermann et al 2013), this is as close as Amery – but on the west side.

Line 217. Delete "most"

Line 223-225 + 229-231: It is not enough to have a deep through, and no polynyas. This is clearly showed by the (overlooked) work from the Fimbul ice shelf, where there is a similar trough and Warm Weddell Deep Water just outside the shelf break (+1.0C). I think there is a major difference in wind forcing that is most essential, there must be little Ekman transport of fresh and cold water towards the SGT. The deep through and no polynyas are also needed – probably – but they are not the only factors.

Line 232 – 237: You do somehow suggest here what I tried to note above, but it is not clear. I think you should compare winds and shore-ward Ekman transport between Fimbul and SGT, and then make a more general and valid conclusion that way.

Line 304: The citation number system creates a strange sentence here, better state: "Fourier transform calculated following Brennan et al (2014).

Figures are generally nice. A few improvements should be done;

Fig. 2. Please draw in a small ice shelf section with depth in all sub figures (just a white bar will do), the red triangle looks weird.

Fig. 4. Some clutter in the labels in the surface layer – please remove.

Fig. 5. Please comment on the discontinuity in the ApRES data, I guess it is year-to-year variability, and you start observing in February? Caption: What is NOFI? Please spell out any abbreviations used the first time.

Fig. 6. Please add the shoreward Ekman transport in the sketch. Maybe it is the fast ice that blocks the wind, and prevents this from being so large as on the Fimbul? Caption: "ice pumped upward transport" is strange. Please use "buoyant melt-water plume". Also use "the thermocline".

Suggested new References:

Galton-Fenzi, B. et al (2012), Modeling the basal melting and marine ice accretion of the Amery Ice Shelf, JGR, 117, C09031, doi:10.1029/2012JC008214.

T. Hattermann et al. Eddy-resolving simulations of the Fimbul Ice Shelf cavity circulation: Basal melting and exchange with open ocean, Ocean Modelling 82 (2014) 28–44

Hatterman, T. (2018), Antarctic thermocline dynamics along a narrow shelf with easterly winds. JPO.

Jenkins, A.: Convection-Driven Melting near the Grounding Lines of Ice Shelves and Tidewater Glaciers, Journal of Physical Oceanography, 41, 2279–2294, <https://doi.org/10.1175/JPO-D-11-03.1>, 2011.

Nøst, O. et al (2011) Eddy overturning of the Antarctic Slope Front controls glacial melting in the Eastern Weddell Sea. *J. Geophys. Res.*, 116, C11 014.

Reviewer #2 (Remarks to the Author):

In this paper, the authors investigate the presence of warm water (classified as modified Circumpolar Deep Water) intruding the sub-ice shelf cavity of Shirase Glacier, a fast-flowing glacier located in East Antarctica. This conjecture is analyzed using past (1990-1992) and more recent hydrographic observations, using simulations from a coupled ocean-sea ice-ice shelf model, and a time series of basal melt rates acquired in situ with an ice radar. The underlying main assumption of this work is the presence of a deep glacial trough in the center of Lützow-Holm Bay connecting to the sub-ice shelf cavity. This topographic feature might constitute a pathway for warm water on the continental slope to infiltrate the sub-ice shelf cavity of Shirase Glacier.

Much of the details for this manuscript are shown in the supplementary material which is really crucial to understand much of the material shown in the main manuscript.

One of the big pieces missing in the main manuscript is a description of the bathymetry. Apart from displaying it in Fig.1a, the authors don't provide any details about how it has been retrieved. From the supplementary material, I was not able to understand whether the bathymetry has been derived out of measurements or it has been reconstructed with a blend of measurements and model. In the latter case, what would be the accuracy?

Having a more comprehensive description of how the authors derived the bathymetry near Shirase Glacier is an extremely important missing detail because measurements (e.g. multibeam) in this sector of Antarctica are rather scarce. If this part of the manuscript is missing, it might be difficult for the reader to fully trust the existence of the topographic features acting as a pathway from the continental shelf to the sub-ice shelf cavity of Shirase Glacier.

Additionally, I also find the conclusions of the manuscript as not comprehensively explained. An average reader might not fully understand what would be the impact of a seasonal (instead of a continuous) basal melt on the global sea level rise, thinning, retreating, and acceleration of the glacier. Please, discuss it more extensively in the conclusions of the manuscript.

Line 49: It might be worth specifying how fast is Shirase Glacier flowing.

Line 183-184: what is the location of the ApRES ice radar measurements with respect to the grounding line of Shirase Glacier? It would be worth to insert the grounding line information in Fig.1a. This would help the reader to understand whether the measurements have been acquired within the grounding zone or far from it.

Line 197: Given that the authors derive the estimated melt rate under several assumption (which are the main ones) and using several estimated parameters, it would be good to quantify the uncertainty on the provided estimates, like it has been done for the satellite-derived case in Line 203. Additionally, through-out the manuscript it is not specified whether the author considered the effect of dynamic thinning on their observed and modelled ice shelf melt rates. Please specify.

Line 250: do you mean 53-98 above the sea floor? This is not clear, please rephrase.

Line 311. Please specify the accuracy and the precision of the ice thickness estimates obtained with the ApRES radar data. In which frequency does the radar operate? What kind of radar is this? FMCW?

Line 314: You mean thermal drift?

Fig 1a. It is extremely difficult to read the labels representing the positions of the CTD stations. The same applies for the contour levels.

Response to the specific comments and suggestions from Reviewer #1

We very much appreciate your constructive and helpful comments. Your comments were invaluable for improving our manuscript. Our response to each comment is written in red italic and we have highlighted the changes made to the revised manuscript with yellow markers. Also, we have highlighted the changes other than those related to the reviewer's comments with light blue markers.

Reviewer #1 (Remarks to the Author):

Antarctica may melt substantially in the future, and contribute largely to future sea level rise. So new melt rates and processes in an area that has not yet been properly observed is indeed interesting for a large audience.

The dynamics that explain the 1 year of seasonal variability is consistent with earlier studies along the larger East Antarctic ice shelf areas. But only comparison is made to the Amery ice shelf to the east. Comparison need also to be done to the Fimbul ice shelf to the west, that is actually more comparable – at least when it comes to the bedrock topography. I also found the “meltwater outflow” index needs both better description, and analysis, before it can be meaningfully applied.

There are some, but not many specific comments below. These go in detail on the larger issues noted above. Otherwise this paper both presents observations, simulations and re-analysis in a clear and well documented story – so I recommend publication after these minor revisions are done.

In the revised manuscript, we could clearly define the characteristics of ice-ocean interaction beneath the Shirase Glacier Tongue through the comparison with the Fimbul Ice Shelf. We greatly appreciate your valuable comment on this issue.

Regarding the “meltwater outflow index”, we have made, and explained, an explicit assumption based on the plume theory by Jenkins (2011).

Also, we have reconstructed our manuscript by incorporating some of the supplementary materials into the main body.

Please see our responses to your specific comments below.

Specific comments:

Generally nice and clear text: “the” is missing in a few places;

Line 85 and 156; “the ice front”, Line 201; “the January value”. Figure 1 caption (Line 475) “the deep through”. Line 540 (Fig. 6 caption) “the thermocline”.

Thank you very much for your careful reading. We have checked and corrected them throughout the manuscript.

Re-analysed wind are essentially a model result, and you need to make the distinction more clear.

Line 152: Add: “Alongshore wind speed from re-analysis on the northeast continental slope ...” It is now written as this is observations. Also change from “wind data” to “wind re-analysis” in Fig. 1 caption (Line 479).

We have changed based on your suggestion. (lines: 155, 617)

Line 118 – 120 is unclear. I suggest re-writing to: “This suggest a circulation pattern with inflow in the sub-surface layer, melting and related buoyant rise below the SGT, and then a northward export of this melt-water product in the surface layer.”

Based on your suggestion, we have modified this part - “This suggests a circulation pattern with along-trough mCDW inflow into the SGT cavity, melting and consequent buoyant rise along the SGT base, and then a northward export of this melt-water product (i.e., a glacial meltwater-mCDW mixture) above the subsurface layer (see the next section for further discussion of the melt-water product).” (line: 119-123)

Line 142-154: In this section only work in this specific part is cited, and tend to be of Japanese authors. The wind driven Ekman transport of the fresher surface layer towards the coast is the same for most of the East Antarctic ice shelves. The paper should at least cite relevant work on the Amery and Fimbul Ice shelf.

We have added a description that the variability in the thermocline depth caused by the easterly wind-driven surface Ekman dynamics, which is common to much of East Antarctica. (line: 157-160)

Line 161 – “Meltwater Outflow Index”. This is not a very clear index. If there is more melting, given the same amount of mixing and flow, the temperature would go down – not up. So here you are assuming that there is more melt when there is also more entrainment – producing the higher temperature. Your assumptions need to be made clear, and you should test if it would be better with simply using

the volume of the melt product only – and leaving the temperature out. What about the thickness of the outflow? This will normally change substantially in an ISW plume (Jenkins 2011). It would also be interesting to learn how much of the variability (Fig. 5 a) is due to speed and temperature.

In reference to the plume theory by Jenkins (2011), we have made an explicit assumption for the definition of this index. (line: 161-173)

As is evident in Fig.2, the upper layer is normally occupied by cold Winter Water with near-freezing temperature, while the melt-water product is observed as a “warm anomaly” at the SGT ice front because of the entrainment of ambient warm modified Circumpolar Deep Water (mCDW) into the meltwater-rich plume. In other words, the northward transport of such a plume would explain the anomalous warm signals observed at the subsurface layer of the SGT ice front. Therefore, the northward current velocity and water temperature in the anomalous subsurface warm layer reflects the magnitude of the overturning circulation accompanied by mCDW entrainment along the SGT base. Further, given the assumption that a plume is well-mixed at the ice shelf base, the basal melt rate can be assumed to depend on the product of the temperature above freezing point ($T - T_f$) and the water speed (v). So, we concluded that it is reasonable to define a “basal melt flux index” as $(T - T_f) \times v$ using subsurface parameters of temperature and northward velocity at ice front strongly associated with the magnitude of basal melting and entrainment processes. For clarity, we have changed the index name from the “meltwater outflow index” to the “basal melt flux index”. Also, variability in this index corresponds well to those in basal melt rates derived from the simulation and in-situ ice radar measurement (Fig.7a, c), and thus, we assume that changes in the index represent those in the basal melt rate across the entire SGT sub-ice cavity.

Line 166-167: Again this is consistent with earlier results along East Antarctica.

Other earlier studies MUST be cited here...

As suggested, we have modified this part by adding other related works here. (line: 181-184)

Line 198-200: The differences between the calculated (25.4 m/yr) and simulated (14.5 m/yr) is quite large, this is OK – but a reasoning around the difference should be made. What is the least robust part of the assumption? Also the year-to-year variability should be discussed shortly. This is clear from the discontinuity in Fig. 5

c) in the ApRES data.

It is difficult to determine which part of the assumption is more robust because this estimate is based on many assumptions such as the outflow area, outflow velocity, and a possible recirculation of the meltwater (see Methods).

However, we acknowledge that the observed $\delta^{18}\text{O}$ value is possibly affected by potential subglacial discharge, and that a scheme to represent subglacial discharge is not implemented in the model. It is possible, therefore, that the oceanographic-derived basal melt rate would be overestimated compared with the results of the simulation. It might be one possible reason for the difference, other than the above, but we decided not to include this discussion in the revised manuscript because the contribution of the subglacial discharge is normally tiny compared with that of the basal melting.

Besides, we have added a brief discussion on the year-to-year variability of ApRES-derived basal melt rate by comparison with the simulated interannual variability shown by one standard deviation of the simulated basal melt rate (Fig.7c). (line: 205-209)

Line 202: should probably be; “the value of simulated” – not the range of simulated

Based on your suggestion, we have changed this part. (line: 224-225)

Line 204; add re-analysis to the data sets used.

We have changed as suggested. (line: 227)

Line 214. Here your results should also be compared with the available melt rates from the Fimbul Ice shelf (Hattermann et al 2014), this is as close as Amery – but on the west side.

In addition to the Amery Ice Shelf, we have added the melt rate comparison with the Fimbul Ice Shelf. (lines: 238-240)

Line 217. Delete “most”

Thank you for pointing out a grammatical error. We have deleted “most” from this part. (line: 243)

Line 223-225 + 229-231: It is not enough to have a deep through, and no polynyas. This is clearly showed by the (overlooked) work from the Fimbul ice shelf, where there is a similar trough and Warm Weddell Deep Water just outside the shelf break (+1.0C). I think there is a major difference in wind forcing that is most essential, there must be little Ekman transport of fresh and cold water towards the SGT. The deep through and no polynyas are also needed – probably – but they are not the only factors.

We have made a substantial revision for this section. (line: 245-273)

By comparing to the Fimbul Ice Shelf located within the same Weddell Gyre system as the SGT, we could clearly define the characteristics of ice-ocean interaction at the SGT and its essential factors- “deep trough”, “permanent cover of land-fast sea ice”, and “absence of active coastal polynyas” in the LH Bay.

Taking into various aspects consideration, we have reached the same conclusion as in the previous manuscript that the most essential local factor is the “continuous deep trough from the outside continental slope to the southernmost tip of the LH Bay” (as shown by the 600 m isobaths, indicated with a thick line in Fig.1a), guiding offshore warm mCDW toward the SGT ice front.

Line 232 – 237: You do somehow suggest here what I tried to note above, but it is not clear. I think you should compare winds and shore-ward Ekman transport between Fimbul and SGT, and then make a more general and valid conclusion that way.

This comment is related to the previous one. Please refer to our previous response. (line: 245-273)

Also, although we have compared the difference in the effect of Ekman dynamics in both regions, we did not include it in the revised manuscript because this context cannot explain why only the SGT has a warm ice cavity as follows- The FIS ice front is located in proximity to the shelf break (Hattermann et al., 2012), while the SGT ice front is located far from the shelf break and the mouth of LHB (Fig.1a). Coupled with the almost permanent cover of heavy land-fast sea ice damping wind stress, the difference in relative ice front positions to the shelf break suggests that the SGT ice cavity is less influenced by Ekman transports of the surface waters, compared to the FIS (e.g., Hattermann et al., 2014). However, this context at least

cannot explain the difference in inflowing water temperatures into the bottom layer of each sub-ice cavity ($> 0^{\circ}\text{C}$ beneath the SGT and up to -1.6°C beneath the FIS).

Line 304: The citation number system creates a strange sentence here, better state: “Fourier transform calculated following Brennan et al (2014).

Based on your suggestion, we have modified this part. (line: 371)

Figures are generally nice. A few improvements should be done;

Fig. 2. Please draw in a small ice shelf section with depth in all sub figures (just a white bar will do), the red triangle looks weird.

As suggested, the SGT has been shown by hatched bar on all subfigures, alternately of the red triangle.

Fig. 4. Some clutter in the labels in the surface layer – please remove.

As suggested, we have reduced the number of labels on this figure (Fig.5).

Fig. 5. Please comment on the discontinuity in the ApRES data, I guess it is year-to-year variability, and you start observing in February?

That’s right, the ApRES was deployed in February 2018 and recovered in January 2019 (observation period: 354 days, see Methods). As you pointed out, the discontinuity in the ApRES data suggests a year-to-year variability.

To support this, we have shown a range of one standard deviation of the simulated basal melt rates that represent an interannual variability (Fig.7c). Also, we have added descriptions on the period of the ApRES data in the caption of Fig.7 (line: 689-690) and a possible interannual variability of the SGT basal melting (line: 205-209).

Caption: What is NOFI? Please spell out any abbreviations used the first time.

In this caption, we have added an explanation of what NOFI is. (line: 682)

Fig. 6. Please add the shoreward Ekman transport in the sketch. Maybe it is the fast ice that blocks the wind, and prevents this from being so large as on the Fimbul? Caption: “ice pumped upward transport” is strange. Please use “buoyant melt-water plume”. Also use “the thermocline”.

As suggested, we have added the onshore Ekman transport in the schematic illustration and changed the caption of Fig.8.

Suggested new References:

Thank you for your suggestion for the new references. We have cited them in the revised manuscript.

Galton-Fenzi, B. et al (2012), Modeling the basal melting and marine ice accretion of the Amery Ice Shelf, JGR, 117, C09031, doi:10.1029/2012JC008214.

T. Hattermann et al. Eddy-resolving simulations of the Fimbul Ice Shelf cavity circulation: Basal melting and exchange with open ocean, Ocean Modelling 82 (2014) 28–44

Hatterman, T. (2018), Antarctic thermocline dynamics along a narrow shelf with easterly winds. JPO.

Jenkins, A.: Convection-Driven Melting near the Grounding Lines of Ice Shelves and Tidewater Glaciers, Journal of Physical Oceanography, 41, 2279–2294, <https://doi.org/10.1175/JPO-D-11-03.1>, 2011.

Nøst, O. et al (2011) Eddy overturning of the Antarctic Slope Front controls glacial melting in the Eastern Weddell Sea. J. Geophys. Res., 116, C11 014.

Response to the specific comments and suggestions from Reviewer #2

We very much appreciate your constructive and helpful comments. Your comments were invaluable for improving our manuscript. Our response to each comment is written in red italic and we have highlighted the changes made to the revised manuscript with yellow markers. Also, we have highlighted the changes other than those related to the reviewer's comments with light blue markers.

Reviewer #2 (Remarks to the Author):

In this paper, the authors investigate the presence of warm water (classified as modified Circumpolar Deep Water) intruding the sub-ice shelf cavity of Shirase Glacier, a fast-flowing glacier located in East Antarctica. This conjecture is analyzed using past (1990-1992) and more recent hydrographic observations, using simulations from a coupled ocean-sea ice-ice shelf model, and a time series of basal melt rates acquired in situ with an ice radar. The underlying main assumption of this work is the presence of a deep glacial trough in the center of Lützow-Holm Bay connecting to the sub-ice shelf cavity. This topographic feature might constitute a pathway for warm water on the continental slope to infiltrate the sub-ice shelf cavity of Shirase Glacier.

Much of the details for this manuscript are shown in the supplementary material which is really crucial to understand much of the material shown in the main manuscript.

We have reconstructed our manuscript by incorporating some of the supplementary materials into the main body.

One of the big pieces missing in the main manuscript is a description of the bathymetry. Apart from displaying it in Fig.1a, the authors don't provide any details about how it has been retrieved. From the supplementary material, I was not able to understand whether the bathymetry has been derived out of measurements or it has been reconstructed with a blend of measurements and model. In the latter case, what would be the accuracy?

Having a more comprehensive description of how the authors derived the bathymetry near Shirase Glacier is an extremely important missing detail because measurements (e.g. multibeam) in this sector of Antarctica are rather scarce. If this part of the manuscript is missing, it might be difficult for the reader to fully trust the existence of the topographic features acting as a pathway from the

continental shelf to the sub-ice shelf cavity of Shirase Glacier.

We agree with your comments since we argue that the presence of the deep trough in Lützow-Holm Bay is the most essential geographical setting for guiding an offshore-origin warm Circumpolar Deep Water toward Shirase Glacier Tongue. So, to provide a proper response to your comments on the bathymetric data in Lützow-Holm Bay, we have added two experts in solid earth geophysics as co-authors (Masakazu Fujii and Yoshifumi Nogi). They have contributed to the acquisition and processing of the bathymetric data used in this manuscript, and they have provided detailed descriptions regarding the bathymetric data.

*We have added a brief and detailed description of the bathymetric data in Methods Section (**line: 285-301**) and Supplementary Information (**new Suppl. S1**), respectively.*

Additionally, I also find the conclusions of the manuscript as not comprehensively explained. An average reader might not fully understand what would be the impact of a seasonal (instead of a continuous) basal melt on the global sea level rise, thinning, retreating, and acceleration of the glacier. Please, discuss it more extensively in the conclusions of the manuscript.

The most striking outcome of our study is the identification of a new hot spot of strong ice-ocean interaction in East Antarctica, induced with an offshore-origin warm water inflow guided by a deep trough continuous from the continental slope.

*Albeit a different perspective from yours, we have added a comprehensive discussion focusing on a close link between the Southern Ocean and Shirase Glacier Tongue (generally applicable to Antarctic Ice Sheet) and its relation to changes in winds and associated ocean surrounding Antarctica in a changing climate. (**line: 245-283**)*

Please see our responses to your specific comments below.

Specific comments:

Line 49: It might be worth specifying how fast is Shirase Glacier flowing.

As suggested, we have added information on the ice velocity of the Shirase Glacier.

(line: 50-52)

Line 183-184: what is the location of the ApRES ice radar measurements with respect to the grounding line of Shirase Glacier? It would be worth to insert the grounding line information in Fig.1a. This would help the reader to understand whether the measurements have been acquired within the grounding zone or far from it.

As suggested, we have added information on the location of the ApRES with respect to the grounding line (from Yamanokuchi et al., 2005), ~16 km north of the SGT grounding line (line: 202). Also, the southernmost position of the SGT grounding line has been superimposed in Fig.1a as the brown line.

Line 197: Given that the authors derive the estimated melt rate under several assumption (which are the main ones) and using several estimated parameters, it would be good to quantify the uncertainty on the provided estimates, like it has been done for the satellite-derived case in Line 203.

Additionally, through-out the manuscript it is not specified whether the author considered the effect of dynamic thinning on their observed and modelled ice shelf melt rates. Please specify.

It is difficult to determine which part of the assumption is more robust because this estimate is based on many assumptions such as the outflow area, outflow velocity, and a possible recirculation of the meltwater (see Methods). Further, it is also difficult to quantify the uncertainty on the estimates because our observation is only a snapshot.

However, we acknowledge that the observed $\delta^{18}\text{O}$ value is possibly affected by potential subglacial discharge, and that a scheme to represent subglacial discharge is not implemented in the model. It is possible, therefore, that the oceanographic-derived basal melt rate would be overestimated compared with the results of the simulation. It might be one possible reason for the difference, other than the above, but we decided not to include this discussion in the revised manuscript because the contribution of the subglacial discharge is normally tiny compared with that of the basal melting.

Also, the ApRES-derived estimates consider the effect of the dynamic thinning of the ice shelf as described in Methods. However, we cannot see what dynamic thinning of the ice shelf has to do with any of the estimates based on the observation or simulation. Thus, we did not include a discussion on how dynamic thinning can have any influence on the basal melt rates derived from observation and simulation in the revised manuscript.

Line 250: do you mean 53-98 above the sea floor? This is not clear, please rephrase.
We have rephrased this part as suggested. (line: 314)

Line 311. Please specify the accuracy and the precision of the ice thickness estimates obtained with the ApRES radar data. In which frequency does the radar operate? What kind of radar is this? FMCW?

Line 314: You mean thermal drift?

In Methods Section, we have added some detailed explanations to provide an answer to your ApRES-related comments. (line: 362-400)

Fig 1a. It is extremely difficult to read the labels representing the positions of the CTD stations. The same applies for the contour levels.

We have modified Fig.1a by improving visualization of the CTD positions and bathymetric contours. The same modification has been applied to Fig.S2.

REVIEWERS' COMMENTS:

Reviewer #2 (Remarks to the Author):

I thank the authors for their thoughtful comments and for their revised manuscript. I think that the paper is now ready for publications.

Our sincere gratitude to two reviewers:

We heartily thank you for your constructive and helpful comments up to this point, which was invaluable for improving our manuscript.

REVIEWERS' COMMENTS:

Reviewer #2 (Remarks to the Author):

I thank the authors for their thoughtful comments and for their revised manuscript. I think that the paper is now ready for publications.